# Poor glycemic control, cardiovascular disease risk factors and their clustering among patients with type 2 diabetes mellitus: A cross-sectional study from Nepal

**Mahesh Kumar Khanal**[1]*, **Pratiksha Bhandari**[2], **Raja Ram Dhungana**[3], **Yadav Gurung**[4], **Lal B. Rawal**[5,6,7], **Gyanendra Pandey**[8], **Madan Bhandari**[1], **Surya Devkota**[9], **Maximilian de Courten**[10], **Barbora de Courten**[11]

1 Provincial Ayurveda Hospital, Ministry of Health, Population and Family Welfare, Dang, Lumbini Province, Nepal, 2 Rapti Life Care Hospital Pvt. Lmt. Tulsipur, Dang, Nepal, 3 Institute for Health and Sport, Victoria University, Melbourne, Australia, 4 Child and Youth Health Research Center, Auckland University of Technology, Auckland, New Zealand, 5 School of Health, Medical and Applied Sciences, College of Science and Sustainability, Central Queensland University, Sydney, Australia, 6 Physical Activity Research Group, Appleton Institute, Central Queensland University, Sydney, Australia, 7 Translational Health Research Institute (THRI), Western Sydney University, Sydney, Australia, 8 Dirghayu Polyclinic and Research Centre, Tulsipur, Dang, Nepal, 9 Department of Cardiology, Manmohan Cardiothoracic Vascular and Transplant Centre, Institute of Medicine, Tribhuvan University, Kathmandu, Nepal, 10 Mitchell Institute for Education and Health Policy, Victoria University, Melbourne, Australia, 11 Department of Medicine, School of Clinical Sciences at Monash Health, Monash University, Clayton, Victoria, Australia

* drmkkhanal@gmail.com

**Data Availability Statement:** All relevant data are within the article and its Supporting Information files.

## Abstract

### Background

Cardiovascular disease (CVD) is the most common complication of diabetes mellitus (DM). To prevent morbidity and mortality among patients with type 2 diabetes mellitus (T2DM), optimization of glycemic status and minimizing CVD risk factors is essential. As Nepal has limited data on these CVD risk parameters, we assessed the prevalence of poor glycemic control, CVD risk factors, and their clustering among patients with T2DM.

### Methods

Using a cross-sectional study design, we collected data of 366 patients with T2DM. We applied a multistage cluster sampling technique and used the WHO STEPS tools. Binary logistic and Poisson regression was applied to calculate odds and prevalence ratio of clustering of risk factors, considering P< 0.05 statistically significant.

### Results

The mean age of participants was 54.5±10.7 years and 208 (57%) were male. The prevalence of poor glycemic control was 66.4% (95% C.I: 61.5–71.2). The prevalence of smoking, alcohol users, inadequate fruit and vegetables intake and physical inactivity were 18% (95% C.I:14 to 21.9), 14.8% (95% C.I:11.1 to 18.4), 98.1% (95% C.I: 96.7–99.4), and 9.8% (95% C.I:6.7–12.8), respectively. Overall, 47.3% (95% C.I: 42.1–52.4) were overweight and

**Funding:** The author(s) received no specific funding for this work.

**Competing interests:** The authors have declared that no competing interests exist.

obese, 59% (95% C.I: 52.9–63) were hypertensive, and 68% (95% C.I: 63.2–72.7) had dyslipidemia. Clustering of two, three, four, five and more than five risk factors was 12.6%, 30%, 30%,19%, and 8.7%, respectively. Four or more risk factors clustering was significantly associated with gender, age, level of education, T2DM duration, and use of medication. Risk factors clustering was significantly higher among males and users of anti-diabetic medications with prevalence ratio of 1.14 (95% C.I:1.05–1.23) and 1.09 (95% C.I: 1.09–1.18)], respectively.

## Conclusions

The majority of the patients with T2DM had poor glycemic control and CVD risk factors. Policies and programs focused on the prevention and better management of T2DM and CVD risk factors should be implemented to reduce mortality in Nepal.

## Background

Diabetes mellitus is a major global public health problem. In the past few decades, the prevalence of diabetes has risen dramatically in all countries and income levels [1]. It was estimated that 451 million adults lived with diabetes worldwide in 2017 [2] and that number is projected to reach 570.9 million by 2025 if effective preventive measures are not adopted [3]. In the latest estimate of 2021, 1.6 million deaths are directly attributed to diabetes each year, and the majority were living in low- and middle-income countries [1]. Diabetes mellitus has an alarming impact on disability-adjusted-life years (DALY) and early deaths. The DALY associated with diabetes in 2017 was 67.9 million and is expected to rise to 79.3 million by 2025 [3]. Similarly, there was a 5% rise in premature life loss between 2000 and 2016 [1].

Cardiovascular disease (CVD), where the heart and blood vessels are negatively impacted, is the number one cause of death in people living with diabetes. CVD accounts for about 60% of all mortality in people with diabetes [4]. One cohort study revealed that compared to patients without diabetes, there were 7.0 (95% CI, 6.7–7.4) and 3.5 (95% CI, 3.3–3.7) deaths/ 1000-person-years higher all-cause and CVD mortality in patients with T2DM, respectively [5]. It is observed that people with diabetes mellitus have a two-fold increase in the risk of stroke in some age groups [4]. Cardiovascular risk increases with poor glucose control [6]. Poor glycemic control corresponds to a higher risk of complications: every 1% increase of glycated haemoglobin (HbA1C) above the threshold level (7%) is associated with a 38% increase in macrovascular events such as CVD [7]. Long-term poor glucose control contributes to the rise of HbA1C level [8]. Furthermore, poor glycemic control along with other risk factors such as central obesity, hypertension, and dyslipidemia contributes to CVD morbidity and mortality [9, 10].

Like in other low- and-middle income countries (LMIC), the prevalence of diabetes mellitus is rising in Nepal. In 2000, only 6.3% of patients had diabetes in a hospital-based study [11]. A population-based survey in 2007 reported that the prevalence of T2DM was 9.5% in semi-urban populations [12]. In 2018, the prevalence of T2DM ranged from 11.7% in the urban population [13] to 16% in the rural population [14]. A nationwide study in 2019 also reported that 8.5% of Nepalese have diabetes mellitus [15]. Additionally, the pooled prevalence of diabetes mellitus from 1990 to 2020 was 8.5% [16]. Even though the burden of T2DM is rising in Nepal, information on glycemic control, CVD risk factors, and their clustering among

patients with T2DM in the sub-urban population of Nepal is limited. Therefore, we aimed to determine the prevalence of poor glycemic control, CVD risk factors, and their clustering among patients with T2DM in the sub-urban population of Nepal.

## Methods

### Study design

The current cross-sectional study was a part of a mixed-method design conducted in the Tulsipur sub-metropolitan city of Dang district of Nepal.

### Study settings

The Tulsipur sub-metropolitan city is one of 16 sub-metropolitan cities of Nepal. In addition, Tulsipur is a famous migration site for people from within and outside the district, such as Salyan, Rukum, and Rolpa in mid-western Nepal. Thus, it gives a heterogeneous population to study. The total population of the city is 1,41,528 [17]. This city has 8 health posts and 11 urban health centers, 2 community health units, 6 hospitals, 2 Ayurvedic hospitals, and 23 polyclinics [17]. Initially, health care facilities were selected randomly. Probable patients were screened and confirmed the diagnosis of T2DM. After that, those patients were invited to visit the Provincial Ayurveda Hospital. This hospital was selected because it is located almost equidistant from all 19 wards (administrative units) of the city. This center is a 30-bedded hospital with specialized services for Ayurveda and complementary treatments that has a basic laboratory setup to analyze blood samples for biochemical tests.

### Study period

The data collection of the cross-sectional study was conducted from December 2021 to February 2022.

### Inclusion and exclusion criteria

We included patients with a confirmed diagnosis of T2DM, who were diagnosed at least one month before data collection, and had visited health facilities of Tulsipur Sub-metropolitan city of Nepal. This T2DM was confirmed from the prescription card of the patient. In our study, mentally ill and pregnant women were excluded.

### Sample size and sampling

The sample size was estimated based on the expected prevalence of physical inactivity (20%) among patients with diabetes in Nepal [18]. Using the formula for estimating proportion for one sample situation with a 5% significance level and a 5% margin of error, the estimated sample size was 245. Our estimated sample size was inflated to 387 to adjust for the design effect (1.5) for multistage cluster sampling and an expected non-response rate of 5%. We used a multistage cluster sampling technique to select representative samples from all kinds of health care facilities. For better coverage, we mobilized community leaders, senior citizens, and health workers in the municipality to encourage patients with T2DM to participate in the study. Out of all health care facilities of Tulsipur Sub-metropolitan City, we randomly selected 4 health posts (50% of total), 4 urban health centers (36% of total), 5 polyclinics (20% of total), 3 hospitals (50% of total) and 1 Ayurvedic hospital (50%) as health care facilities for recruiting patients.

## Data collection

We recruited and trained local health professionals and medical students in data collection tools and techniques as enumerators. These enumerators were responsible for obtaining written informed consent from the participants. With the patient's approval, we gathered information related to socio-demographic, behavioral, and anthropometric data using WHO STEPS questionnaires. Then, participants were invited to visit Provincial Ayurveda Hospital in the early morning with at least eight hours of fasting for the blood test. In the laboratory, about 2 ml of blood was withdrawn for fasting blood sugar, fasting lipid profile, and creatinine measurement. Again, patients were requested to provide blood for postprandial blood sugar testing after two hours of their usual meal. We tried to maximize the participation to provide blood sample by calling and recalling the patients with T2DM on their mobile or landline phones, if someone was missed in the first attempt. All biochemical assessments were performed on a semi-automated machine at no cost to the patient. During the study period, person-to-person counselling was provided based on the latest reports and behavioral and anthropometric status. Finally, we asked patients to consult a physician or make regular visits to their usual health care facilities for follow-up.

## Study variables

**Demographic and socio-economic variables.** We considered completed years of age. We categorized the level of education as illiterate, primary (grade 1 to 5), secondary (grade 6–10), and more than secondary (>10 grade). Based on caste, patients were re-categorized into Brahman, Kshetri, Janajati (Chaudhary, Magar, Gurung, and other indigenous castes), and Dalit. On the basis of occupation related data, participants were categorized into employed, self-employed (running their own business or farming), household work, and unemployed (student, non-paid worker, or retired). For economic status, participants were defined as poor if household income per person per year was less than 19,262 rupee (US $158) [19].

**Behavior-related variables (STEP I).** For behavior-related information, current smokers were defined as participants who had smoked at least one month prior to the data collection. Current non-smoking tobacco users were labeled as those who consumed tobacco apart from smoking. Current alcohol users were defined as those who consumed alcoholic beverages within the last month. We measured fruit and vegetable intake using a pictogram. Sufficient fruit and vegetable intake was defined as consumption of at least five servings of fruit and vegetables per day (400 gm). Similarly, we also asked about the type of cooking oil. Plant-based oil was considered if participants consumed mustard or sunflower oil. Furthermore, we measured physical activity in metabolic equivalents of tasks (METs) minutes per week. A low level of physical activity was defined as less than 600 METs minutes per week of physical activity [20].

**Anthropometric variables (STEP II).** A stadiometer was used to measure height at the nearest 0.1 centimeters. Similarly, a weighing machine was used to measure weight to the nearest 0.1 kilograms (kg). Participants were classified as underweight (<18.5 kg/m2), normal (18.5–24.9 kg/m2), overweight (25–29.9 kg/m2), or obese (> 30 kg/m2) based on their BMI [21]. We used a non-stretchable measuring tape to measure waist circumference (WC) at the level of the umbilicus in the non-inspiration phase, to the nearest 0.1 cm. Waist circumference (central obesity) was considered as raised if the female participant had more than 88 cm of waist circumference and the male participant had more than 102 cm [22]. We measured two blood pressure readings in the sitting position in the left arm using an aneroid sphygmomanometer to the nearest 2 mmHg, first at least 15 minutes rest and second 3 minutes after the first reading. For the final analysis, we took the average of two readings. Hypertension was defined as an average systolic blood pressure (SBP) $\geq$ 140 mmHg and/or an average diastolic

blood pressure (DBP) ≥ 90 mmHg and/or a history of taking antihypertensive medication in the last 2 weeks [23].

**Biochemical variables.** *Poor glycemic control*. Poor glycemic control was defined by the American Diabetes Association as fasting blood sugar levels greater than 130 mg/dl and/or postprandial blood sugar levels greater than 180 mg/dl [8].

*Dyslipidemia*. To determine dyslipidemia, we measured fasting lipid profile. Dyslipidemia was defined as having at least one of the following: high total cholesterol (≥200 mg/dl), high triglyceride (≥150 mg/dl), high low-density lipoprotein (130 mg/dl), low high-density lipoprotein (≤40 mg/dl in males and ≤50 mg/dl in females), and/or use of antilipidemic drugs [24]. These lipid profiles were assessed in the laboratory, except for low-density lipoprotein (LDL) which was calculated using the Friedewald formula [25].

**Clustering of poor glycemic control and cardiovascular risk factors.** The clustering of modifiable CVD risk factors was assessed based on the presence of eight major risk factors; poor glycemic control, current smoking, current alcohol use, inadequate intake of fruit and vegetables, low physical activity, overweight or obesity, hypertension, and dyslipidemia [14, 26].

## Validity and pre-testing of data collection tools

The STEPS questionnaires had previously been tested, translated, and used [14] in Nepal. For the current study, we had pre-tested the data collection tools among 15 samples by trained interviewers and technicians. On completion of pre-testing, feedback and comments were taken. Finally, we adapted to the feedback and refined the tool.

## Data analysis

We compiled, edited, and checked the collected data to maintain consistency. We omitted repetitions and corrected them before coding and entering them in EpiData version 3.1. Then, the statistician exported the data to SPSS V.20.0 for further analysis. Out of 366 data, 11 related to blood samples were missing. We used expectation maximization (EM) to impute the missing data using age, duration of diabetes, history of oral hypoglycemic drug intake, blood sugar level, lipid profile, and creatinine. We used simple descriptive statistics to demonstrate socio-demographic characteristics. For continuous variables, we used mean and standard deviation (SD). We calculated the prevalence of categorical individual CVD risk factors as a percentage with a 95% confidence interval (CI). We separately calculated all the risk factors and added them individually to determine the clustering of risk factors in percentage. Chi-square and independent t-tests were used to compare normally distributed categorical and continuous variables, respectively. For binary logistic regression, we recoded clustering of poor glycemic control and CVD risk factors as yes (1): if clustering of four or more risk factors and/or poor glycemic control were present, and no (0): if fewer than four risk factors were present. This recoding of CVD risk factor clustering was done as more than half of the respondents had four to seven risk factors clustering. Socio-demographic and clinical variables were entered into binary and multivariable logistic regression models to determine crude and adjusted odds ratios of CVD risk factors clustering, respectively. Additionally, we performed Poisson regression to see the effect of those variables on the clustering of one to eight risk factors. We used a two-tail test. We considered $p < 0.05$ to be statistically significant.

## Ethical consideration

The Ethical Review Board (ERB) of the Nepal Health Research Council reviewed the ethical issues, research proposal, and endorsed the final version of the protocol (reference number

1430). We prepared a written consent form in the Nepali language. During implementation, our data enumerators read the form and described the study objectives, procedures of data collection, risks and benefits of the study, and the confidentiality of their personal information. All participants signed or provided a thumb impression (if unable to write) on a separate consent form before enrolling in the study. Current study did not include minors.

### Inclusivity in global research

Additional information regarding the ethical, cultural, and scientific considerations specific to inclusivity in global research is included in the Supporting Information (S1 Checklist).

## Results

### Socio-demographic and clinical characteristics of study participants

Out of 366, males were 56.8% and females were 43.2%. The mean age of study participants was 54.5 years with a standard deviation (SD) of 10.7 years. The majority (64.8%) were in the age range of 46–65 years. Regarding education, most of the participants had either primary (31%) or secondary (31%) education. However, among females, 39% were illiterate and 39% studied until grade five or less. Most of the study patients were married (95.4%). Out of all, 45% were household workers. However, 34.6% of males were running their own businesses. Twenty-one percent of the participants were poor (Table 1).

The mean duration of T2DM was 4.4 years (±4.9 years). Men had a significantly longer duration of diabetes mellitus compared to women (4.9±5.4 years vs. 3.8±4.1 years, p = 0.04). Almost two-thirds of patients (71%) were taking oral anti-diabetic medicine. Overall, 2.5% were taking both insulin and an oral hypoglycemic drug. Of all, one-third (35%) had a history of diabetes mellitus in their first-degree relatives (Table 2).

### Poor glycemic control among study participants

The prevalence of poor glycemic control was 66.4% (95% C.I: 61.5–71.2) among the study population. More than one-third (36%) were male and less than one-third (30%) were female (Table 3). Overall, the mean FBS was 135.6±60.3 mg/dl and the mean PPBS was 218.2±81.6 mg/dl (Table 2). The highest prevalence of poor glycemic control was among participants in the age group 46 to 65 years old (45.4%), having primary level education (19%), doing household work, and being above the poverty line (52.7%). However, there was no significant difference among these groups (Table 3).

### CVD risk factors of study participants

**Behavioral risk factors.** The result shows that 18% of all participants were currently smoking (95% C.I: 14–21.9). The prevalence of current smokers was significantly higher in men compared to women (16% vs. 1.6%, p<0.001). Current smoking was significantly associated with the level of education, with the highest prevalence among participants having a secondary education (p = 0.003). Similarly, 17.2% (95% C.I: 13.3–21) were currently using non-smoking tobacco. A significantly higher proportion of males were using tobacco during the study period compared to females (15.8% vs. 1.4%, p<001). The prevalence of current alcohol users was 14.8% (95% C.I: 11.1–18.4) with more men (13.9%) compared to women (0.8%, p<001) (Tables 3 and 4). Almost all participants were consuming less than five servings of fruit and vegetables 98.1% (95% C.I: 96.7–99.4). Overall, the mean serving of fruit and vegetable intake was 2.1±1. About one-tenth of participants were consuming non-plant-based

**Table 1. Distribution of socio-demographic characteristics of study participants by gender (N = 366).**

| Variables | | Total n (%) | Male n (%), N = 208 | Female n (%), N = 158 | P value |
|---|---|---|---|---|---|
| **Age (years)** | | | | | |
| | 26–45 | 74 (20.2) | 33 (15.9) | 41 (26) | 0.039 |
| | 46–65 | 237 (64.8) | 139 (66.8) | 98 (62) | |
| | >65 | 55 (15) | 36 (17.3) | 19 (12) | |
| **Level of education** | | | | | |
| | Illiterate | 88 (24) | 25 (12) | 63 (39.9) | **<0.001** |
| | Primary | 112 (30.6) | 50 (24) | 62 (39.2) | |
| | Secondary | 113 (30.9) | 89 (42.8) | 24 (15.2) | |
| | Higher | 53 (14.5) | 44 (21.2) | 9 (5.7) | |
| **Ethnicity** | | | | | |
| | Bhramhan | 106 (29) | 63 (30.3) | 43 (27.2) | 0.51 |
| | Kshetri | 185 (50.5) | 108 (51.9) | 77 (48.7) | |
| | Janajati | 44 (12) | 21 (10.1) | 23 (14.6) | |
| | Dalit | 31 (8.5) | 16 (7.7) | 15 (9.5) | |
| **Marital status** | | | | | |
| | Married | 349 (95.4) | 199 (95.7) | 150 (94.9) | 0.74 |
| | Unmarried | 17 (4.6) | 9 (4.3) | 8 (5.1) | |
| **Occupation** | | | | | |
| | Employed | 49 (13.4) | 34 (16.3) | 15 (9.5) | **<0.001** |
| | Self-employed | 102 (29.7) | 72 (34.6) | 30 (19) | |
| | Household-work | 165 (45.1) | 60 (28.8) | 105 (66.5) | |
| | Unemployed | 50 (13.7) | 42 (20.2) | 8 (5.1) | |
| **Economic status** | | | | | |
| | Below poverty line* | 78 (21.3) | 48 (23.1) | 30 (19) | 0.34 |
| | Above poverty line | 288 (78.8) | 160 (76.9) | 128 (81) | |

*Household income per person per year less than 19,262 rupee (US $158)

vegetable oil (9%). The prevalence of physical inactivity was 9.8% (95% C.I: 6.7–12.8). The mean MET was 5769 ±4302 minutes/week. (Tables 2 and 3).

**Anthropometric risk factors.** The prevalence of overweight and obesity was 47.3% (95% C.I: 42.1–52.4). About one-tenth of participants were obese, and 37% were overweight. On analyzing gender, 20% of men were overweight, whereas 5.7% of women were obese. The mean BMI was 25±3.6 kg/m$^2$ with women having a slightly higher BMI (25.5±3.6 kg/m2) than men (24.5±3.6 kg/m2, p = 0.01) and obesity was associated with age (p<0.001). The prevalence of obesity was highest among those aged 46 to 64 (32.5%). Similarly, 39.6% (95 C.I: 34.5–44.6) had an increased waist circumference. Women had central obesity at a rate nearly three times higher (p<0.001) than men (9%). The mean waist circumference was 89.5±18.6 cm among all participants. Central obesity was significantly associated with the level of education (p<0.001) and occupation (p = 0.002). Participants having secondary education had a higher prevalence of elevated waist circumference (21.6%). In the same way, household workers had a higher prevalence of elevated WC (23.2%) (Tables 2–4).

**Hypertension.** The prevalence of hypertension was 59% (95% C.I: 52.9–63.0). Thirty-four percent of men and 25% of women had hypertension. Overall, the mean SBP was 130.4±16.9 mmHg with 131.7±16.5 mmHg in males and 128.7±17.2 mmHg in females. The mean DBP was 85.1±10.7 mmHg in the whole group, with 85.7±11 mmHg in men and 84.3±10.3 mmHg in women. Hypertension was associated with age (p<0.001), level of education (p = 0.03), and

**Table 2. Clinical characteristics of study participants stratified by gender (N = 366).**

| Variables | | Total | Male | Female | P value |
|---|---|---|---|---|---|
| **Age (years)** | Mean (SD) | 54.5 (10.7) | 55.9 (10.7) | 52.5 (10.5) | **0.003** |
| **Duration of diabetes (years)** | | | | | |
| | Mean (SD) | 4.4 (4.9) | 4.9 (5.4) | 3.8 (4.1) | **0.04** |
| Up to 1 year | n (%) | 116 (31.7) | 65 (17.8) | 51 (13.9) | 0.16 |
| >1 to-5 years | n (%) | 149 (40.7) | 78 (21.3) | 71 (19.4) | |
| >5 years | n (%) | 101 (27.6) | 65 (17.8) | 36 (9.8) | |
| **Anti-diabetic medication** | | | | | |
| No | n (%) | 106 (29) | 57 (15.6) | 49 (13.4) | 0.45 |
| Oral | n (%) | 260 (71) | 151 (41.3) | 109 (29.8) | |
| Oral+Insuline | n (%) | 9 (2.5) | 6 (1.6 | 3 (0.8) | |
| **Family history of diabetes** | | | | | |
| Yes | n (%) | 129 (35.2) | 77 (21) | 52 (14.2) | 0.41 |
| No | n (%) | 237 (64.8) | 131 (35.8) | 106 (29) | |
| **FV (servings/day)** | Mean (SD) | 2.1 (1.0) | 2.08 (0.94) | 2.16 (1.15) | 0.46 |
| **MET (minutes/week)** | Mean (SD) | 5769.1 (4302.1) | 6007.4 (4292.7) | 5455 (4307.9) | 0.22 |
| **BMI(Kg/m2)** | Mean (SD) | 25.0 (3.6) | 24.5 (3.6) | 25.5 (3.6) | **0.01** |
| **WC (cm)** | Mean (SD) | 89.5 (18.6) | 90.2 (17.7) | 88.6 (19.8) | 0.42 |
| **SBP (mmHg)** | Mean (SD) | 130.4 (16.9) | 131.7 (16.5) | 128.7 (17.2) | 0.09 |
| **DBP (mmHg)** | Mean (SD) | 85.1 (10.7) | 85.7 (11.0) | 84.3 (10.3) | 0.22 |
| **FBS (mg/dl)** | Mean (SD) | 135.6 (60.3) | 134.3 (64.3) | 137.2 (54.0) | 0.64 |
| **PPBS (mg/dl)** | Mean (SD) | 218.2 (81.6) | 216.8 (86.2) | 220.2 (75.4) | 0.69 |
| **TC (mg/dl)** | Mean (SD) | 152.8 (39.1) | 150.4 (38.6) | 156.0 (39.6) | 0.17 |
| **TG (mg/dl)** | Mean (SD) | 173.0 (102.4) | 172.8 (98.2) | 173.2 (107.9) | 0.97 |
| **HDL (mg/dl)** | Mean (SD) | 43.8 (7.3) | 41.2 (6.4) | 47.3 (7.1) | **<0.001** |
| **LDL (mg/dl)** | Mean (SD) | 74.3 (3.5) | 74.6 (34.6) | 74.0 (35.7) | 0.88 |

FV (Servings of fruit and/or vegetable), METs (Metabolic equivalents of tasks), BMI (Body Mass Index), WC (Waist circumference), SBP (Systolic Blood Pressure), DBP (Diastolic Blood Pressure), FBS (Fasting Blood Sugar), PPBS (Post-prandial Blood Sugar), TC (Total Cholesterol), TG (Triglyceride), HDL (High-density Lipoprotein), LDL (Low-density Lipoprotein)

economic status (p = 0.02). The participants aged 46 to 65 years, having a secondary education, and having economic status above the poverty line (38.8%, 18%, and 44%, respectively) had the highest prevalence of hypertension (Tables 2–4).

**Dyslipidemia.** The prevalence of dyslipidemia was 68% (95% C.I: 63.2–72.7). This prevalence was significantly higher in males compared to females (42.6% vs. 25.4%, p = 0.001). Out of all, 42.6% of men and 25.4% of women had dyslipidemia. The mean total cholesterol was 152.8±39.1 mg/dl. The mean triglyceride, HDL, and LDL were 173±102.4 mg/dl, 43.8±7.3 mg/dl, and 74.3±3.5 mg/dl, respectively. Only the mean HDL was significantly different between males and females (p<0.001) (Tables 2 and 3).

## Clustering of poor glycemic control and CVD risk factors

All participants had either poor glycemic control or other CVD risk factors. Overall, 12.6% of patients with T2DM had a clustering of two risk factors, with a higher prevalence in women (14.6%) compared to men (6.3%, p = 0.001). Sixty percent of participants had a clustering of either three or four risk factors. Women had a higher proportion of clustering of three and four risk factors (36% and 31%) compared to men (25% and 28.8%), respectively (p = 0.001). Almost one-fifth (19%) had five risk factors and 8.7% had more than five risk factors

**Table 3. Prevalence of poor glycemic control and other cardiovascular disease risk factors among study participants stratified by gender (N = 366).**

| Variables | | Total | | Gender | | |
|---|---|---|---|---|---|---|
| | | n (%) | 95% C.I. | Male n (%) | Female n (%) | P value |
| **Poor glycemic control** | | | | | | |
| | Yes | 243 (66.4) | 61.5–71.2 | 133 (36.3) | 110 (30.1) | 0.25 |
| | No | 123 (33.6) | | 75 (20.5) | 48 (13.1) | |
| **Behavioral CVD risk factors** | | | | | | |
| **Current smoker** | | | | | | |
| | Yes | 66 (18) | 14.0–21.9 | 60 (16.4) | 6 (1.6) | **<0.001** |
| | No | 300 (82) | | 148 (40.4) | 152 (41.5) | |
| **Current tobacco user** | | | | | | |
| | Yes | 63 (17.2) | 13.3–21.0 | 58 (15.8) | 5 (1.4) | **<0.001** |
| | No | 303 (82.8) | | 150 (41) | 153 (41.8) | |
| **Current alcohol user** | | | | | | |
| | Yes | 54 (14.8) | 11.1–18.4 | 51 (13.9) | 3 (0.8) | **<0.001** |
| | No | 312 (85.2) | | 157 (42.9) | 155 (42.3) | |
| **Fruit and/or vegetable intake** | | | | | | |
| | Inadequate | 359 (98.1) | 96.7–99.4 | 205 (56) | 154 (42.1) | 0.45 |
| | Adequate | 7 (1.9) | | 3 (0.8) | 4 (1.1) | |
| **Cooking oil** | | | | | | |
| | Non plant based oil | 33 (9) | 6.0–11.9 | 21 (5.6) | 12 (3.2) | 0.40 |
| | Plant based oil | 333 (91) | | 187 (51.1) | 146 (39.9) | |
| **Physical activity** | | | | | | |
| | Inactive | 36 (9.8) | 6.7–12.8 | 16 (4.4) | 20 (5.5) | 0.11 |
| | Active | 330 (90.2) | | 192 (52.5) | 138 (37.7) | |
| **Anthropometric and clinical risk factors** | | | | | | |
| **Body Mass Index** | | | | | | |
| | Underweight | 8 (2.2) | | 5 (1.4) | 3 (0.8) | 0.16 |
| | Normal | 185 (50.5) | | 114 (31.1) | 71 (19.4) | |
| | Overweight | 136 (37.2) | 32.2–42.1 | 73 (19.9) | 63 (17.2) | |
| | Obese | 37 (10.1) | 7.0–13.1 | 16 (4.4) | 21 (5.7) | |
| **Waist circumference** | | | | | | |
| | Raised | 145 (39.6) | 34.5–44.6 | 33 (9) | 112 (30.6) | **<0.001** |
| | Normal | 221 (60.4) | | 175 (47.8) | 46 (12.6) | |
| **Hypertension** | | | | | | |
| | Yes | 216 (59) | 52.9–63.0 | 125 (34.2) | 91 (24.9) | 0.63 |
| | No | 150 (41) | | 83 (22.7) | 67 (18.3) | |
| **Dyslipidemia** | | | | | | |
| | Yes | 249 (68) | 63.2–72.7 | 156 (42.6) | 93 (25.4) | **0.001** |
| | No | 117 (32) | | 52 (14.2) | 65 (17.8) | |

clustering. However, the prevalence of five or more risk factors clustering was higher in men than in women (13.5% vs. 2.5%, p = 0.001) (Fig 1).

Clustering of four or more CVD risk factors, including poor glycemic control, was significantly associated with gender, age group, level of education, ethnicity, duration of T2DM, and use of oral medications (Table 5). The odds of risk factors clustering among men was 1.18 (95% C.I: 1.19–2.77) times higher than that among women. After adjustment for covariates, this odds ratio (OR) increased to 2.04 (95% C.I: 1.18–3.54). After adjustment for covariates, the odds of clustering of four or more risk factors were 1.94 (95% C.I: 1.01–3.72) more among

**Table 4. Distribution of poor glycemic control and CVD risk factors among study participants stratified by age and socio-economic characteristics.**

| Variables | Poor Glycemic control | Hypertension | Dyslipidemia | Overweight and obesity | Central obesity | Smoking | Alcohol user | Inadequate Fruit/ vegetable intake | Physical Inactivity |
|---|---|---|---|---|---|---|---|---|---|
| **Age (years)** | | | | | | | | | |
| 26–45 | 12 | 8.5 | 12.8 | 11.5 | 9.8 | 3 | 4.1 | 19.9 | 1.6 |
| 46–65 | 45.4 | 38.8 | 45.4 | 32.5 | 39.9 | 12.8 | 10.4 | 63.1 | 5.7 |
| >65 | 9 | 11.7 | 9.8 | 3.3 | 10.7 | 2.2 | 0.3 | 15 | 2.5 |
| *P value* | *0.13* | *<**0.001*** | *0.52* | *<**0.001*** | *0.31* | *0.47* | ***0.009*** | *0.53** | *0.2* |
| **Level of education** | | | | | | | | | |
| Illiterate | 17.2 | 15.3 | 14.2 | 9.6 | 12.8 | 2.7 | 2.2 | 23.5 | 2.5 |
| Primary | 19.1 | 19.7 | 20.5 | 17.5 | 14.8 | 4.1 | 3.8 | 30.1 | 3.6 |
| Secondary | 2.8 | 18 | 21.9 | 14.2 | 21.6 | 9 | 6.6 | 30.6 | 3.6 |
| Higher | 9.3 | 6 | 11.5 | 6 | 11.2 | 2.2 | 2.2 | 13.9 | 0.3 |
| *P value* | *0.57* | ***0.03*** | *0.08* | *0.06* | *<**0.001*** | ***0.003*** | *0.09** | *0.64** | *0.2* |
| **Ethnicity** | | | | | | | | | |
| Bhramhan | 17.8 | 16.4 | 21 | 11.2 | 18 | 3.6 | 2.2 | 27.9 | 3 |
| Kshetri | 33.9 | 30.6 | 34.4 | 27.6 | 29.8 | 10.4 | 8.8 | 49.7 | 4.9 |
| Janajati | 8.5 | 5.7 | 7.4 | 4.1 | 7.4 | 2.5 | 3.3 | 12 | 1.1 |
| Dalit | 6.3 | 6.3 | 5.2 | 4.4 | 5.2 | 1.6 | 1.1 | 8.5 | 0.8 |
| *P value* | *0.48* | *0.12* | *0.46* | *0.15* | *0.95* | *0.33* | *0.16** | *0.32** | *0.99** |
| **Marital status** | | | | | | | | | |
| Unmarried | 3.6 | 2.2 | 4.1 | 2.2 | 3 | 0.8 | 1.1 | 4.6 | 0 |
| Married | 62.8 | 56.8 | 63.9 | 45.1 | 57.4 | 17.2 | 13.7 | 93.4 | 9.8 |
| *P value* | *0.36* | *0.3* | *0.06* | *0.98* | *0.7* | *0.96** | *0.29** | *0.55** | *0.16** |
| **Occupation** | | | | | | | | | |
| Employed | 7.7 | 7.1 | 10.1 | 6.3 | 10.1 | 1.4 | 1.9 | 13.1 | 0.8 |
| Self-employed | 17.2 | 17.8 | 18.3 | 12.6 | 16.7 | 4.9 | 4.6 | 27.3 | 2.7 |
| Household-work | 32.8 | 26.5 | 29.8 | 22.7 | 23.2 | 7.9 | 4.9 | 44.3 | 4.6 |
| Unemployed | 8.7 | 7.7 | 9.8 | 5.7 | 10.4 | 3.8 | 3.3 | 13.4 | 1.6 |
| *P value* | *0.114* | *0.6* | *0.53* | *0.71* | ***0.002*** | *0.14* | *0.13* | *0.99** | *0.78** |
| **Economic status** | | | | | | | | | |
| Below poverty line | 13.7 | 15 | 12.8 | 9.6 | 13.4 | 3.3 | 2.5 | 21 | 2.7 |
| Above poverty line | 52.7 | 44 | 55.2 | 37.7 | 47 | 14.8 | 12.3 | 77 | 7.1 |
| *P value* | *0.62* | ***0.02*** | *0.09* | *0.63* | *0.62* | *0.49* | *0.36* | *0.64** | *0.31* |

* Fischer's exact

the age cohort of 46–65 years compared to the age group older than 65 years. The odds of clustering of risk factors was 2.22 (95% C.I: 1.13–4.28) times higher among patients with secondary education than among those with higher education. Every year of T2DM duration increased the risk factor clustering by an odds ratio of 1.05 (95% C.I: 1–1.09). The odds of clustering of risk factors was 1.71 (95% CI 1.03–2.84) times greater among participants taking anti-diabetic medications compared to patients not taking any medications, even after adjusting for co-variates (Table 5).

In Poisson regression, after adjusting for age, level of education, ethnicity, marital status, occupation, economic status, duration of diabetes, and history of taking an oral medication, the chance of clustering was 14% higher in males compared to females with an adjusted

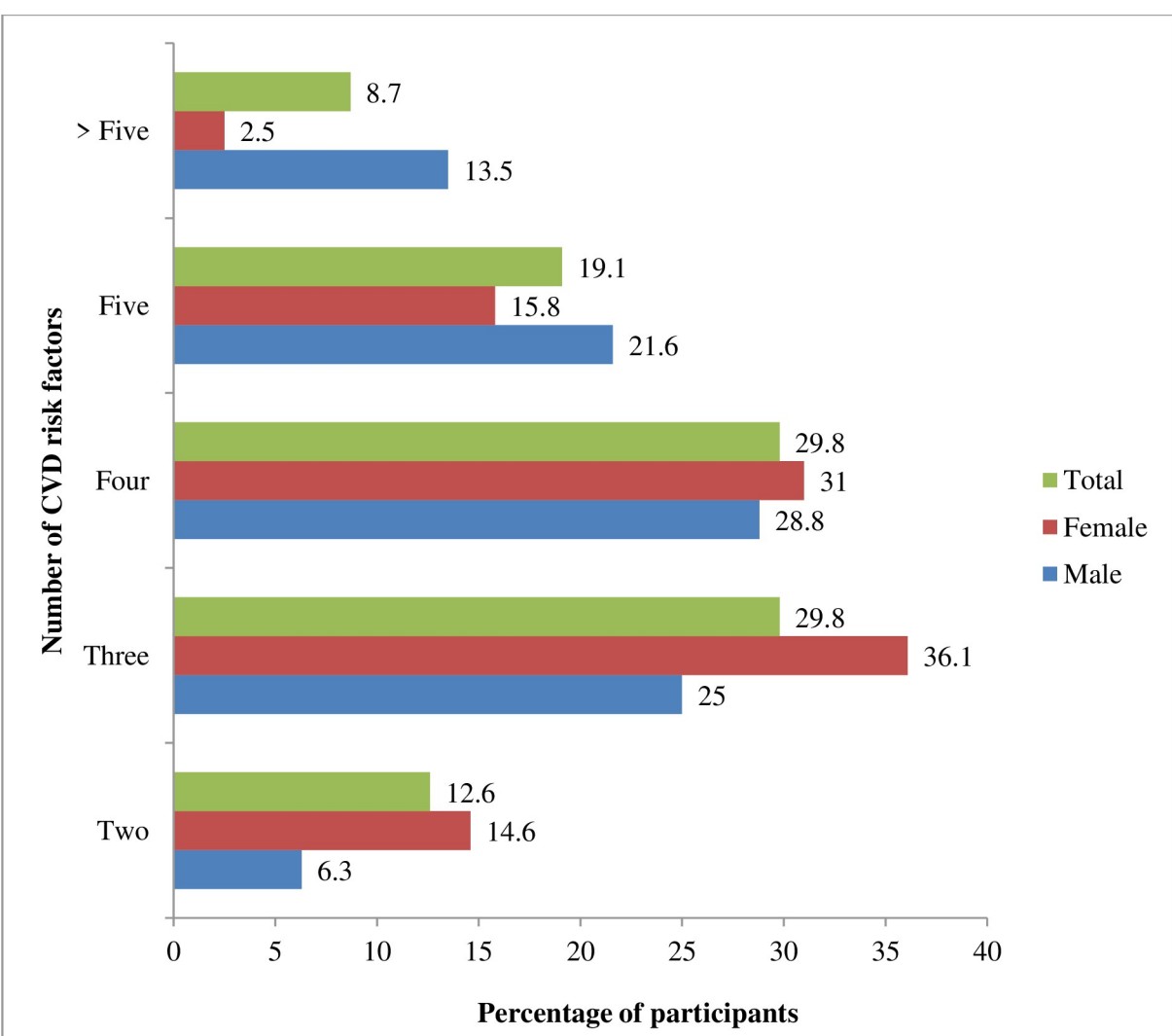

**Fig 1. Clustering of CVD risk factors among patients with T2DM.**

prevalence ratio of 1.14 (95% C.I. 1.05–1.23). Similarly, after adjusting for all other covariates, patients taking oral hypoglycemic medications were 9% more likely to have a clustering of risk factors than those not on medication with a PR of 1.09 (95% C.I: 1.01–1.18) (Table 6).

## Discussion

The current study determined the prevalence of poor glycemic control, CVD risk factors, and their clustering among patients with T2DM in Tulsipur Sub-metropolitan city of Nepal. Almost two-thirds of patients had uncontrolled blood sugar levels either in fasting or post-prandial blood. A significant percentage of study participants had behavioral, anthropometric, and intermediary risk factors for CVD. Almost sixty percent had three or four risk factors for CVD, and more than half had four or more risk factors. This clustering was more common among men and those who were taking hypoglycemic medication.

One out of three patients with T2DM had poor glycemic control, similar to a previous study by Thakur et al. in Nepal, which reported that 70% of diabetes patients had poor glucose

**Table 5. Socio-demographic and clinical factors associated with clustering four or more cardiovascular disease risk factors among study participants.**

| Variables | | Crude OR (95% CI) | Adjusted OR (95% CI) |
|---|---|---|---|
| **Gender** | | | |
| | Female | Reference | |
| | Male | 1.18 (1.19–2.77)* | 2.04 (1.18–3.54)* |
| **Age (years)** | | | |
| | 26–45 | 1.09 (0.54–2.20) | 1.58 (0.69–3.61) |
| | 46–65 | 1.66 (0.92–3.00) | 1.94 (1.01–3.72)* |
| | >65 | Reference | |
| **Level of education** | | | |
| | Illiterate | 1.38 (0.69–2.74) | 2.04 (0.83–4.99) |
| | Primary | 1.79 (0.93–3.48) | 2.30 (1.00–5.22)* |
| | Secondary | 2.22 (1.13–4.28)* | 2.19 (1.02–4.68)* |
| | Higher | Reference | |
| **Ethnicity** | | | |
| | Bhramhan | Reference | |
| | Kshetri | 1.51 (0.93–2.44) | 1.20 (0.69–2.09) |
| | Janajati | 0.96 (0.47–1.94) | 0.86 (0.39–1.91) |
| | Dalit | 2.35 (0.99–5.58)* | 1.95 (0.73–5.18) |
| **Marital status** | | | |
| | Unmarried | Reference | |
| | Married | 1.22 (0.46–3.24) | 1.20 (0.41–3.50) |
| **Occupation** | | | |
| | Employed | Reference | |
| | Self-employed | 1.42 (0.71–2.83) | 1.13 (0.52–2.47) |
| | Household-work | 1.24 (0.65–2.34) | 1.13 (0.53–3.23) |
| | Unemployed | 1.70 (0.76–3.81) | 1.31 (0.53–3.23) |
| **Economic status** | | | |
| | Below poverty line | Reference | |
| | Above poverty line | 1.06 (0.64–1.76) | 1.25 (0.70–2.23) |
| **Duration of Diabetes** | | 1.05 (1.00–1.09)* | 1.03 (0.98–1.08) |
| **Oral medication** | | | |
| | No | Reference | |
| | Yes | 1.92 (1.21–3.03)* | 1.71 (1.03–2.84)* |

*Significant at level <0.05

control [27]. This result is also consistent with other studies in another developing country, Ethiopia, reporting a prevalence of 68.3% [28] and 70.9% [29]. However, poor glycemic control in India was higher than in our study (76.6% to 77.6%) [30, 31]. The current study's high prevalence of poor glycemic control could be explained by several factors, including failure to follow a diabetic diet, insufficient physical activity, and failure to take adequate medications. For example, about 99% of participants were not eating sufficient fruits and vegetables. Similarly, 10% of patients were physically inactive, and 30% were not taking their medications. Out of those who were taking their medications, there might have been inadequate dosing, insufficient drug combinations, or non-adherence to the treatment [32, 33]. Previous studies have demonstrated that individual and health system-related factors significantly maintain reasonable glycemic control. These are lack of knowledge about self-management practices, poor economic status, inaccessible health services, unhealthy lifestyle (unhealthy diet and physical

**Table 6. Determinants of clustering of CVD risk factors (Poisson regression analysis).**

| Variables | | Mean of risk factors clustering | Adjusted prevalence ratio (95% C.I.) |
|---|---|---|---|
| **Gender** | | | |
| | Female | 3.38 | Reference |
| | Male | 3.86 | 1.14 (1.05–1.23)* |
| **Age (years)** | | | |
| | 26–45 | 3.61 | 1.04 (0.92–1.18) |
| | 46–65 | 3.79 | 1.09 (0.99–1.21) |
| | >65 | 3.45 | Reference |
| **Level of education** | | | |
| | Illiterate | 3.58 | 1.05 (0.91–1.20) |
| | Primary | 3.74 | 1.10 (0.97–1.24) |
| | Secondary | 3.75 | 1.10 (0.98–1.23) |
| | Higher | 3.4 | Reference |
| **Ethnicity** | | | |
| | Bhramhan | 3.43 | Reference |
| | Kshetri | 3.68 | 1.07 (0.98–1.17) |
| | Janajati | 3.58 | 1.04 (0.92–1.17) |
| | Dalit | 3.78 | 1.10 (0.96–1.26) |
| **Marital status** | | | |
| | Unmarried | 3.73 | Reference |
| | Married | 3.5 | 1.06 (0.91–1.24) |
| **Occupation** | | | |
| | Employed | 3.51 | Reference |
| | Self-employed | 3.57 | 1.01 (0.90–1.14) |
| | Household-work | 3.72 | 1.06 (0.94–1.18) |
| | Unemployed | 3.66 | 1.04 (0.91–1.18) |
| **Economic status** | | | |
| | Below poverty line | 3.56 | Reference |
| | Above poverty line | 3.67 | 1.03 (0.94–1.12) |
| **Duration of Diabetes** | | | 1.00 (0.99–1.01) |
| **Oral medication** | | | |
| | No | 3.45 | Reference |
| | Yes | 3.78 | 1.09 (1.01–1.18)* |

*Significant at level <0.05

inactivity), medication-related counseling, lack of motivation from family and peers, missing specific guidelines for medical practitioners, and lack of training and motivation of health professionals–all of which could be barriers to good glycemic control in patients with T2DM in Nepal [34–36].

Regarding behavioral risk factors of CVD, 17.2% of our study population smoked tobacco. Smoking prevalence in other studies of Nepal ranged from 12% [27] to 27% [37] among patients with diabetes. Our findings also correlate with the results of a nationwide STEPS survey of the general population in Nepal in 2019, which determined a 17% of prevalence of smoking [38]. However, the current smoking in our study is higher than in the study of patients with diabetes in India (9.6%) [31]. Similarly, the prevalence of alcohol users among patients with T2DM in our study was approximately 15%, slightly lower than that found in the aforementioned national study (20%) on the general population [38] but slightly higher than

that of India (10.2%) [31]. In our study, two-thirds of the participants were Kshetri, Janajati, and Dalit, who had a higher chance of lifetime alcohol consumption [39]. The prevalence of physically inactive diabetic patients was 9.8% in our study, which is half (20%) of another study conducted in an urban setting in Nepal [18]. However, 38.4% of patients with T2DM were physically inactive in Saudi Arabia, four times higher than that in the current study [40]. The lower prevalence of physical inactivity in our study might be due to the enrolment of participants from semi-urban settings, with the majority being employed. In our study, almost all of the participants ate less than five servings of fruit and vegetables (98%). This is comparable to a nationally representative sample of the general population in Nepal (97%) [38]. That means semi-urban patients with T2DM were not different in their fruit and vegetable consumption.

Almost half of our participants were overweight or obese. This is much lower than the prevalence determined by a hospital-based study in Nepal, which reported that raised BMI and WC were 71.2% and 86%, respectively [37]. Similarly, the current prevalence of obesity is far less than that of another study (88.3%) conducted among patients with T2DM in Nepal [27]. Similarly, comparing our results with findings from India, with a prevalence ranging from 68.4% [31] to 87% [30], the current prevalence of overweight and obesity is lower. The present study's lower prevalence of overweight and obesity would be due to higher physical activity. A majority of patients with T2DM in our study (90%) were physically active.

The prevalence of hypertension in our study was 59%. This is higher than other studies conducted in tertiary hospitals in Nepal, which reported that 28% and 41% had raised blood pressure [37, 41]. Similarly, a tertiary hospital-based study in India revealed that 37.8% of patients with T2DM had hypertension [31]. The lower prevalence of hypertension in the previous studies [37, 41] (conducted among tertiary hospitals) compared to the current study in a suburban area could be explained by the study population. This higher gap in hypertension in the current study site indicates a higher future burden of CVD in the regions where tertiary care services are inaccessible.

More than two-thirds of patients with T2DM in the current study had dyslipidemia. The current prevalence of dyslipidemia is also comparable with other studies, which reported that 61% [42] to 88% [37, 43, 44] of Nepalese patients with T2DM had dyslipidemia. Similarly, the prevalence of dyslipidemia in patients with T2DM in India ranged from 34.7% [31] to more than 85% [45]. The high prevalence of dyslipidemia in the current study is likely due to poor diet and obesity, and it alleged that there could be no regular lipid measurements and monitoring performed in a sub-urban setting.

One striking finding of the current study is that every diabetic patient had at least one risk factor for CVD. More than half of the study participants had four to eight CVD risk factors, including poor glycemic control. A similar prevalence was reported in a study by Hussein et al., which determined that 16%, 37%, 24.5% and 22.5% had one, two, three, and four or more risk factors clustering of CVD among patients with T2DM [46]. The clustering of four or more risk factors in the current study is associated with gender, age, level of education, ethnicity, duration of diabetes, and use of hypoglycemic oral medications. Therefore, there needs to be a special focus on middle-aged men with less than secondary level education and patients with a longer duration of diabetes, including those on oral hypoglycemic therapy. Our results demonstrated that even taking oral medicine for T2DM is not enough to reduce the risk factors of CVD, which could be due to a lack of compliance and medical follow-up to optimize the treatment in the rural area. Although there are not enough reported results on clustering of CVD to compare with our findings, we have found that most patients with T2DM have several CVD risk factors. Therefore, any improvement inT2DM control in Nepal should be through a comprehensive chronic disease management model rather than pursuing disease-specific

models of care [47–49] to minimize the clustering of CVD risk. Diabetes mellitus should be a key component of a noncommunicable disease (NCD) prevention and control action plan [50], which includes easily accessible and quality health services, trained and motivated health professionals, and regular patient follow-up. In resource-constrained settings, such as rural and sub-urban regions of Nepal, where there is a shortage of physicians, trained non-physician mid-level practitioners can help with T2DM care [51]. Similarly, evidence showed that counseling, care coordination in communities, and follow-up would be effective through female community health volunteers (FCHV) [52]. If patients do not have the knowledge and understand the chronicity of the condition and the importance of diabetes self-care, they will not follow lifestyle modifications and seek healthcare advice and service. Future research should focus on how to provide affordable, accessible, and effective diabetes care in a chronic disease management program involving physicians, nurses, dieticians, and community health care workers in order to improve diabetes care and reduce CVD risk factors.

There are a few limitations to the current study. First, we could not measure glycated hemoglobin (HbA1C) because of the unavailability and cost of the test. HbA1C provides the average level of blood sugar over two to three months. Fasting or postprandial blood sugar might overestimate or underestimate the accurate picture of glycemic control among patients with T2DM. Similarly, we were not able to assess the food frequency and consumption of a diabetic diet. Fruit and vegetable intake is not enough to assess the impact of diet on glycemic control. In addition, we could not measure medication compliance among the patients. Additionally, the sample size of the current study was small. Although the current study site of Tulsipur Sub-metropolitan City represents the majority of sub-urban areas of Nepal, a study in a single community cannot represent the case of the whole country. Therefore, our findings must be generalized cautiously.

## Conclusions

Two out of three patients with T2DM had poor glycemic control, and all of the participants had at least one CVD risk factor. The most common additional CVD risk factors were dyslipidemia, hypertension, overweight or obesity, and inadequate fruit and vegetable intake. Male patients and people treated with oral hypoglycemic medications were more likely to have a higher number of risk factors clustering. Therefore, health care improvement and new models of care are imminent to improve diabetes control and minimize the CVD risk factor profile of patients with T2DM to prevent the future burden of cardiovascular diseases.

## Supporting information

**S1 Checklist. STROBE Statement—Checklist of items that should be included in reports of *cross-sectional studies.***
(DOC)

**S1 File. Inclusivity in global research.**
(DOCX)

**S1 Dataset.**
(SAV)

## Acknowledgments

We want to acknowledge tremendous help from Pratik Bhandari, a PhD candidate from Deakin University, Australia, for providing technical support and language correction. Authors

are thankful to Provincial Ayurveda Hospital Development Committee for providing laboratory reagents to conduct biochemical tests. We are grateful for the relentless fieldwork by supervisors- Dr. Ashish Kumar Jha, Dr. Pradeep Adhikari, Dr. Rakesh Yadav, and Dr. Bijay Kishor Thakur. Data collection would not be completed without support from senior volunteers; Dr. Bikash Rijal, Dr. Dharma Raj Chaudhary, Dr. Mahendra Khadka, Dr. Gita Shahu, Dr. Trishmita Chaudhary, Dr, Reshu Budhathoki, Dr. Sushma Chaudhary, Dr. Sumitra Rana, Dr. Shikhar Pokhrel, Dr. Krishna Singh Dhami, Dr. Abha Adhikari, Dr. Manoj Yadav, Ms Parbati Oli, Ms. Lalita Dangi, and Ms. Sonu D. C. Additionally, we want to thank junior volunteers; Mr. Umesh Tharu, Ms. Sweta Kafle, Ms. Urmila Sharma, Ms. Sushmita Shrestha, Ms. Sunita Sunar, Mr. Sanjib Tharu, Ms. Samjhana Dhodari, Ms. Samjhana Tharu, Ms. Kriti Singh Chaudhary, Ms. Ashma Budhathoki, Mr. Samir Rawol, Ms. Sailbala Sharma, Mr. Rita Kuwar, Mr. Rajaram Upadhyay, Mr. Kapil Neupane, Mr. Keshav Khanal, Rabina Chaudhary, Mr. Jeewan Sharma, and Mr. Mitra Bhusal for contributing in data collection. We cannot forget valuable contribution and effort given by laboratory staff of Provincial Ayurveda hospitals; Mrs. Hira Pokhrel, Mr. Rohit K. C., Ms. Dilmaya Gharti, Ms. Pratiksha Oli. Our research implementation would not be finished without moral support from all Tulsipur Sub-metropolitan city health workers. We want to extend our thanks to the head of the department of health of Tulsipur Sub-metropolitan city Mr. Om P. Neupane, and other senior staff, Mr. Bimal K.C. Additionally, we want to thank Mr. Dorna Oli,Jiban Bhandari and Hari Rijal of Rapti Life Care Hospital, Mr. Jaye Prakash Oli of Dirghayu Polyclinic, ward chairmen of respective wards, Diabetic Society of Tulsipur for supporting the implementation of the research project. In the end, we are grateful to the participants of the study.

## Author Contributions

**Conceptualization:** Mahesh Kumar Khanal, Pratiksha Bhandari, Raja Ram Dhungana, Yadav Gurung, Lal B. Rawal, Gyanendra Pandey, Madan Bhandari, Surya Devkota, Maximilian de Courten, Barbora de Courten.

**Data curation:** Mahesh Kumar Khanal, Pratiksha Bhandari, Raja Ram Dhungana, Yadav Gurung, Gyanendra Pandey, Madan Bhandari, Surya Devkota, Maximilian de Courten, Barbora de Courten.

**Formal analysis:** Mahesh Kumar Khanal, Pratiksha Bhandari, Raja Ram Dhungana, Yadav Gurung, Lal B. Rawal, Gyanendra Pandey, Maximilian de Courten, Barbora de Courten.

**Investigation:** Raja Ram Dhungana, Gyanendra Pandey.

**Methodology:** Mahesh Kumar Khanal, Pratiksha Bhandari, Raja Ram Dhungana, Yadav Gurung, Gyanendra Pandey, Madan Bhandari, Surya Devkota, Maximilian de Courten, Barbora de Courten.

**Project administration:** Pratiksha Bhandari, Gyanendra Pandey, Madan Bhandari.

**Resources:** Yadav Gurung, Lal B. Rawal, Madan Bhandari, Surya Devkota, Barbora de Courten.

**Software:** Yadav Gurung.

**Supervision:** Mahesh Kumar Khanal, Pratiksha Bhandari, Yadav Gurung, Lal B. Rawal, Gyanendra Pandey, Madan Bhandari, Maximilian de Courten, Barbora de Courten.

**Validation:** Raja Ram Dhungana, Lal B. Rawal, Madan Bhandari, Surya Devkota, Maximilian de Courten, Barbora de Courten.

**Visualization:** Pratiksha Bhandari, Raja Ram Dhungana, Yadav Gurung, Lal B. Rawal.

**Writing – original draft:** Mahesh Kumar Khanal.

**Writing – review & editing:** Mahesh Kumar Khanal, Pratiksha Bhandari, Raja Ram Dhungana, Yadav Gurung, Lal B. Rawal, Gyanendra Pandey, Madan Bhandari, Surya Devkota, Maximilian de Courten, Barbora de Courten.

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
