## [Decision Letter · Decision Letter 0]

8 Jun 2022

PONE-D-22-05557Poor glycemic control, cardiovascular disease risk factors and their clustering among patients with type 2 diabetes mellitus: A cross-sectional study from NepalPLOS ONE

Dear Dr. Khanal,

Thank you for submitting your manuscript to PLOS ONE. After careful consideration, we feel that it has merit but does not fully meet PLOS ONE’s publication criteria as it currently stands. Therefore, we invite you to submit a revised version of the manuscript that addresses the points raised during the review process.

We look forward to receiving your revised manuscript.

Kind regards,

Rick J. Jansen, PhD, MS

Academic Editor

PLOS ONE

Journal Requirements:

3. Please include a complete copy of PLOS’ questionnaire on inclusivity in global research in your revised manuscript. Our policy for research in this area aims to improve transparency in the reporting of research performed outside of researchers’ own country or community. The policy applies to researchers who have travelled to a different country to conduct research, research with Indigenous populations or their lands, and research on cultural artefacts. The questionnaire can also be requested at the journal’s discretion for any other submissions, even if these conditions are not met.  Please find more information on the policy and a link to download a blank copy of the questionnaire here: https://journals.plos.org/plosone/s/best-practices-in-research-reporting. Please upload a completed version of your questionnaire as Supporting Information when you resubmit your manuscript.

Reviewers' comments:

Reviewer's Responses to Questions

**Comments to the Author**

1. Is the manuscript technically sound, and do the data support the conclusions?

Reviewer #1: Yes

Reviewer #2: Yes

2. Has the statistical analysis been performed appropriately and rigorously? 

Reviewer #1: Yes

Reviewer #2: Yes

3. Have the authors made all data underlying the findings in their manuscript fully available?

Reviewer #1: Yes

Reviewer #2: Yes

4. Is the manuscript presented in an intelligible fashion and written in standard English?

Reviewer #1: Yes

Reviewer #2: No

5. Review Comments to the Author

Reviewer #1: The authors examined the prevalence of poor glycemic control, CVD risk factors, and their clustering among patients with T2DM and concluded that the majority of the patients with T2DM had poor glycemic control and CVD

risk factors in the city Tulsipur in Nepal. Needs for the policies and programs focused on the prevention and better management of T2DM and CVD risk factors were rised.

This is a very well conducted study and written paper on a very important topic.

I do not see any major flaw.

Reviewer #2: I have gone through the manuscript and found it well drafted. The scientific rigor and other technical issues are generally acceptable based on my experience in the field.

However, I need some clarifications or rebuttal for the following points:

Abstract: Method

• Poisson logistic regression vs Poisson regression: Which one is generally acceptable to use?

Introduction,

• Instead of using the term ‘recent years’, ‘latest’ etc., I recommend the researchers to provide the exact dates/years for readers several years later. Update this section with latest citations as well

Methods

• Include study period in the subtitle (rename as ‘study design, settings and period’)

• …….confirmed diagnosis of T2DM, who were diagnosed at least one month before data collection……Why did you use one month as time limit?

• “……T2DM was confirmed from the prescription card of the patient. ………..We selected patients with T2DM from randomly selected centers and then they were invited to visit the Provincial Ayurveda Hospital….”. Which one was applied initially? You contacted the patient based on the card as tracer or contacted patient then back to the card for confirmation. Clarify

• Sample size and sampling � sample size determination and sampling procedure

• It that a simple or multistage cluster sampling?

• In simple cluster sampling, all units in the selected cluster should be studied, whereas in multistage, double sampling procedures for settings and individuals should be done.

• Page 7: line 124 ……to select representative samples from all kinds of health centers � rewrite as health facilities

• After collecting socio-demographic, anthropometric and, related variables, you invited the patient to visit the hospital with eight hours of fasting. What did you do when the patient missed/overlooked your invitation?

• …….We asked patients to provide blood for postprandial blood sugar testing after two hours of their usual meal…….This might be tiresome for the patients as this was the third time to contact investigators. This part would have been collected when you collected the sociodemo….related variables just to measure Random blood sugar. If the intention was to measure oral glucose tolerance test (OGTT), the investigator him/herself should provide oral glucose (fixed dose) two hours earlier.

• Mixed up of definition of terms with description of variables

• “We defined poor glycemic control as fasting blood sugar >130 milligrams per deciliter (mg/dl) and/or postprandial blood sugar ≥180 mg/dl [8]”. Many studies available online,however, used a cut-off point 126 mg/dl for FBS and 200 mg/dl for RBS. Provide a justification for it.

• The biochemical variables should be clearly defined again with clear cut-off points and in line with the universal standard (needs separate operational definitions)

• Page 10: Line 211-212: ….yes (1) if clustering of four or more risk factors and/or poor glycemic control were present, and no (0): if fewer than three risk factors were present….. What about three?

• Page 10: Line 214-215: Socio-demographic and clinical variables were entered into simple and multiple logistic regression models to determine crude and adjusted odds ratios of clustering of CVD risk factors, respectively. Simple and multiple logistic regression � Bivariable and multivariable BINARY logistic regression? Clarify if it is to mean different from this.

Generally,

• Scientific procedures in method section shall be preferentially written in passive voice. Almost all the entire manuscript was written in active voice.

• The active voice is direct, clear and concise and the passive voice is indirect, but sometimes it is useful to emphasize the research, instead of the researcher in scientific world.

• I left this part to the discretion of the research to provide justification.

• I recommend the researchers to use the balance between the two forms.

6. PLOS authors have the option to publish the peer review history of their article (what does this mean?). If published, this will include your full peer review and any attached files.

Reviewer #1: No

Reviewer #2: **Yes: **Mekonnen Sisay

---

## [Author Response · Author response to Decision Letter 0]

17 Jun 2022

Response to both the reviewers have been separately uploaded in documents file.

---

## [Editor Report · Decision Letter 1]

11 Jul 2022

Poor glycemic control, cardiovascular disease risk factors and their clustering among patients with type 2 diabetes mellitus: A cross-sectional study from Nepal

PONE-D-22-05557R1

Dear Dr. Khanal,

We’re pleased to inform you that your manuscript has been judged scientifically suitable for publication and will be formally accepted for publication once it meets all outstanding technical requirements.

Kind regards,

Rick J. Jansen, PhD, MS

Academic Editor

PLOS ONE
---

## [Editor Report · Acceptance letter]

14 Jul 2022

PONE-D-22-05557R1 

Poor glycemic control, cardiovascular disease risk factors and their clustering among patients with type 2 diabetes mellitus: A cross-sectional study from Nepal 

Dear Dr. Khanal:

I'm pleased to inform you that your manuscript has been deemed suitable for publication in PLOS ONE. Congratulations! Your manuscript is now with our production department. 

Kind regards, 

on behalf of

Dr. Rick J. Jansen 

Academic Editor

PLOS ONE